# Gradient-Guided Importance Sampling for Learning Binary Energy-Based Models

**Meng Liu, Haoran Liu, Shuiwang Ji**
Department of Computer Science & Engineering
Texas A&M University
College Station, TX 77843, USA
`{mengliu,liuhr99,sji}@tamu.edu`

## Abstract

Learning energy-based models (EBMs) is known to be difficult especially on discrete data where gradient-based learning strategies cannot be applied directly. Although ratio matching is a sound method to learn discrete EBMs, it suffers from expensive computation and excessive memory requirements, thereby resulting in difficulties in learning EBMs on high-dimensional data. Motivated by these limitations, in this study, we propose ratio matching with gradient-guided importance sampling (RMwGGIS). Particularly, we use the gradient of the energy function *w.r.t.* the discrete data space to approximately construct the provably optimal proposal distribution, which is subsequently used by importance sampling to efficiently estimate the original ratio matching objective. We perform experiments on density modeling over synthetic discrete data, graph generation, and training Ising models to evaluate our proposed method. The experimental results demonstrate that our method can significantly alleviate the limitations of ratio matching, perform more effectively in practice, and scale to high-dimensional problems. Our implementation is available at `https://github.com/divelab/RMwGGIS`.

## 1 Introduction

Energy-Based models (EBMs), also known as unnormalized probabilistic models, model distributions by associating unnormalized probability densities. Such methods have been developed for decades (Hopfield, 1982; Ackley et al., 1985; Cipra, 1987; Dayan et al., 1995; Zhu et al., 1998; Hinton, 2012) and are unified as energy-based models (EBMs) (LeCun et al., 2006) in the machine learning community. EBMs have great simplicity and flexibility since energy functions are not required to integrate or sum to one, thus enabling the usage of various energy functions. In practice, given different data types, we can parameterize the energy function with different neural networks as needed, such as multi-layer perceptrons (MLPs), convolutional neural networks (CNNs) (LeCun et al., 1998), and graph neural networks (GNNs) (Gori et al., 2005; Scarselli et al., 2008). Recently, EBMs have been drawing increasing attention and are demonstrated to be effective in various domains, including images (Ngiam et al., 2011; Xie et al., 2016; Du & Mordatch, 2019), videos (Xie et al., 2017), texts (Deng et al., 2020), 3D objects (Xie et al., 2018), molecules (Liu et al., 2021; Hataya et al., 2021), and proteins (Du et al., 2020b).

Nonetheless, learning (*a.k.a.*, training) EBMs is known to be challenging since we cannot compute the exact likelihood due to the intractable normalization constant. As reviewed in Section 4, many approaches have been proposed to learn EBMs, such as maximum likelihood training with MCMC sampling (Hinton, 2002) and score matching (Hyvärinen & Dayan, 2005). However, most recent advanced methods cannot be applied to discrete data directly since they usually leverage gradients over the continuous data space. For example, for many methods based on maximum likelihood training with MCMC sampling, they use the gradient *w.r.t.* the data space to update samples in each MCMC step. However, if we update discrete samples using such gradient, the resulting samples are usually invalid in the discrete space. Therefore, learning EBMs on discrete data remains challenging.

Ratio matching (Hyvärinen, 2007) is a method to learn discrete EBMs on binary data by matching ratios of probabilities between the data distribution and the model distribution, as detailed in Sec-

tion 2.2. However, as analyzed in Section 3.1, it requires expensive computations and excessive memory usages, which is infeasible if the data is high-dimensional.

In this work, we propose to use the gradient of the energy function *w.r.t.* the discrete data space to guide the importance sampling for estimating the original ratio matching objective. More specifically, we use such gradient to approximately construct the provably optimal proposal distribution for importance sampling. Thus, the proposed approach is termed as ratio matching with gradient-guided importance sampling (RMwGGIS). Our RMwGGIS can significantly overcome the limitations of ratio matching. In addition, it is demonstrated to be more effective than the original ratio matching in practice. We perform extensive analysis for this improvement by connecting it with hard negative mining, and further propose an advanced version of RMwGGIS accordingly by reconsidering the importance weights. Experimental results on synthetic discrete data, graph generation, and Ising model training demonstrate that our RMwGGIS significantly alleviates the limitations of ratio matching, achieves better performance with obvious margins, and has the ability of scaling to high-dimensional relevant problems.

## 2 PRELIMINARIES

### 2.1 ENERGY-BASED MODELS

Let $\boldsymbol{x}$ be a data point and $E_{\boldsymbol{\theta}}(\boldsymbol{x}) \in \mathbb{R}$ be the corresponding energy, where $\boldsymbol{\theta}$ represents the learnable parameters of the parameterized energy function $E_{\boldsymbol{\theta}}(\cdot)$. The probability density function of the model distribution is given as $p_{\boldsymbol{\theta}}(\boldsymbol{x}) = \frac{e^{-E_{\boldsymbol{\theta}}(\boldsymbol{x})}}{Z_{\boldsymbol{\theta}}} \propto e^{-E_{\boldsymbol{\theta}}(\boldsymbol{x})}$, where $Z_{\boldsymbol{\theta}} \in \mathbb{R}$ is the normalization constant (*a.k.a.*, partition function). To be specific, $Z_{\boldsymbol{\theta}} = \int e^{-E_{\boldsymbol{\theta}}(\boldsymbol{x})} d\boldsymbol{x}$ if $\boldsymbol{x}$ is in the continuous space and $Z_{\boldsymbol{\theta}} = \sum e^{-E_{\boldsymbol{\theta}}(\boldsymbol{x})}$ for discrete data. Hence, computing $Z_{\boldsymbol{\theta}}$ is usually infeasible due to the intractable integral or summation. Note that $Z_{\boldsymbol{\theta}}$ is a variable depending on $\boldsymbol{\theta}$ but a constant *w.r.t.* $\boldsymbol{x}$.

### 2.2 RATIO MATCHING

Ratio matching (Hyvärinen, 2007) is developed for learning EBMs on binary discrete data by matching ratios of probabilities between the data distribution and the model distribution. Note that we focus on $d$-dimensional binary discrete data $\boldsymbol{x} \in \{0, 1\}^d$ in this work.

Specifically, ratio matching considers the ratio of $p(\boldsymbol{x})$ and $p(\boldsymbol{x}_{-i})$, where $\boldsymbol{x}_{-i} = (x_1, x_2, \cdots, \bar{x}_i, \cdots, x_d)$ denotes a point in the data space obtained by flipping the $i$-th dimension of $\boldsymbol{x}$. The key idea is to force the ratios $\frac{p_{\boldsymbol{\theta}}(\boldsymbol{x})}{p_{\boldsymbol{\theta}}(\boldsymbol{x}_{-i})}$ defined by the model distribution $p_{\boldsymbol{\theta}}$ to be as close as possible to the ratios $\frac{p_{\mathcal{D}}(\boldsymbol{x})}{p_{\mathcal{D}}(\boldsymbol{x}_{-i})}$ given by the data distribution $p_{\mathcal{D}}$. The benefit of considering ratios of probabilities is that they do not involve the intractable normalization constant $Z_{\boldsymbol{\theta}}$ since $\frac{p_{\boldsymbol{\theta}}(\boldsymbol{x})}{p_{\boldsymbol{\theta}}(\boldsymbol{x}_{-i})} = \frac{e^{-E_{\boldsymbol{\theta}}(\boldsymbol{x})}}{Z_{\boldsymbol{\theta}}} \cdot \frac{Z_{\boldsymbol{\theta}}}{e^{-E_{\boldsymbol{\theta}}(\boldsymbol{x}_{-i})}} = e^{E_{\boldsymbol{\theta}}(\boldsymbol{x}_{-i}) - E_{\boldsymbol{\theta}}(\boldsymbol{x})}$. To achieve the match between ratios, Hyvärinen (2007) proposes to minimize the objective function

$$\mathcal{J}_{RM}(\boldsymbol{\theta}) = \mathbb{E}_{\boldsymbol{x} \sim p_{\mathcal{D}}(\boldsymbol{x})} \sum_{i=1}^{d} \left[ g\left( \frac{p_{\mathcal{D}}(\boldsymbol{x})}{p_{\mathcal{D}}(\boldsymbol{x}_{-i})} \right) - g\left( \frac{p_{\boldsymbol{\theta}}(\boldsymbol{x})}{p_{\boldsymbol{\theta}}(\boldsymbol{x}_{-i})} \right) \right]^2 + \left[ g\left( \frac{p_{\mathcal{D}}(\boldsymbol{x}_{-i})}{p_{\mathcal{D}}(\boldsymbol{x})} \right) - g\left( \frac{p_{\boldsymbol{\theta}}(\boldsymbol{x}_{-i})}{p_{\boldsymbol{\theta}}(\boldsymbol{x})} \right) \right]^2 .$$

$$(1)$$

The sum of two square distances with the role of $\boldsymbol{x}$ and $\boldsymbol{x}_{-i}$ switched is specifically designed since it is essential for the following simplification. In addition, the function $g(u) = \frac{1}{1+u}$ is also carefully chosen in order to obtain the subsequent simplification. To compute the objective defined in Eq. (1), it is known that the expectation over data distribution (*i.e.*, $\mathbb{E}_{\boldsymbol{x} \sim p_{\mathcal{D}}(\boldsymbol{x})}$) can be unbiasedly estimated by the empirical mean of samples $\boldsymbol{x} \sim p_{\mathcal{D}}(\boldsymbol{x})$. However, to obtain the ratios between $p_{\mathcal{D}}(\boldsymbol{x})$ and $p_{\mathcal{D}}(\boldsymbol{x}_{-i})$ in Eq. (1), the exact data distribution is required to be known, which is usually impossible.

Fortunately, thanks to the above carefully designed objective, Hyvärinen (2007) demostrates that the objective function in Eq. (1) is equivalent to the following simplified version

$$\mathcal{J}_{RM}(\boldsymbol{\theta}) = \mathbb{E}_{\boldsymbol{x} \sim p_{\mathcal{D}}(\boldsymbol{x})} \sum_{i=1}^{d} \left[ g\left( \frac{p_{\boldsymbol{\theta}}(\boldsymbol{x})}{p_{\boldsymbol{\theta}}(\boldsymbol{x}_{-i})} \right) \right]^2 \tag{2}$$

which does not require the data distribution to be known and can be easily computed by evaluating the energy of $\boldsymbol{x}$ and $\boldsymbol{x}_{-i}$, according to $\frac{p_{\boldsymbol{\theta}}(\boldsymbol{x})}{p_{\boldsymbol{\theta}}(\boldsymbol{x}_{-i})} = e^{E_{\boldsymbol{\theta}}(\boldsymbol{x}_{-i}) - E_{\boldsymbol{\theta}}(\boldsymbol{x})}$. Further, Lyu (2009) shows that the above objective function of ratio matching agree with the following objective

$$\mathcal{J}_{RM}(\boldsymbol{\theta}) = \mathbb{E}_{\boldsymbol{x} \sim p_{\mathcal{D}}(\boldsymbol{x})} \sum_{i=1}^{d} \left[ \frac{p_{\boldsymbol{\theta}}(\boldsymbol{x}_{-i})}{p_{\boldsymbol{\theta}}(\boldsymbol{x})} \right]^2 = \mathbb{E}_{\boldsymbol{x} \sim p_{\mathcal{D}}(\boldsymbol{x})} \sum_{i=1}^{d} \left[ e^{E_{\boldsymbol{\theta}}(\boldsymbol{x}) - E_{\boldsymbol{\theta}}(\boldsymbol{x}_{-i})} \right]^2 . \tag{3}$$

They agree with each other since function $g(\cdot)$ decreases monotonically in $[0, +\infty)$, which aligns with the value range of probability ratios $\frac{p_{\boldsymbol{\theta}}(\boldsymbol{x})}{p_{\boldsymbol{\theta}}(\boldsymbol{x}_{-i})}$. Note that Eq. (3) is originally named as the *discrete extension of generalized score matching* in Lyu (2009). Since it agrees with Eq. (2) and includes ratios of probabilities, we also treat it as an objective function of ratio matching in our context. In the rest of this paper, we provide our analysis and develop our method based on Eq. (3) for clarity. Nonetheless, our proposed method below can be naturally performed on Eq. (2) as well.

Intuitively, the objective function of ratio matching, as formulated in Eq. (3) or Eq. (2), can push down the energy of the training sample $\boldsymbol{x}$ and push up the energies of other data points obtained by flipping one dimension of $\boldsymbol{x}$. Thus, this objective faithfully expect that each training sample $\boldsymbol{x}$ has higher probability than its local neighboring points that are hamming distance 1 from $\boldsymbol{x}$.

It is worth mentioning that ratio matching is not suitable for binary image data. This is because it faithfully treats $\boldsymbol{x}$ as a positive sample and $\boldsymbol{x}_{-i}$ for $i = 1, 2, \cdots, d$ as negative samples. However, this assumption does not hold for binary image data. Specifically, if we flip one dimension/pixel of a binary image such as a digit image from MNIST, the resulting image is almost unchanged and is still a positive sample in nature. Thus, it is not reasonable to use ratio matching to train EBMs on binary images. Similarly, our RMwGGIS presented below also has this limitation. Having said this, our RMwGGIS is still an effective and efficient method for a broad of binary discrete data. We leave extending its applicability to general discrete EBMs for future work.

## 3 THE PROPOSED METHOD

We first analyze the limitations of the ratio matching method from the perspective of computational time and memory usage. Then, we describe our proposed method, ratio matching with gradient-guided importance sampling (RMwGGIS), which uses the gradient of the energy function *w.r.t.* the discrete input $\boldsymbol{x}$ to guide the importance sampling for estimating the original ratio matching objective. Our approach can alleviate the limitations significantly and is shown to be more effective in practice.

### 3.1 ANALYSIS OF RATIO MATCHING

**Time-intensive computations.** Based on Eq. (3), for a sample $\boldsymbol{x}$, we have to compute the energies for all $\boldsymbol{x}_{-i}$, where $i = 1, \cdots, d$. This needs $\mathcal{O}(d)$ evaluations of the energy function for each training sample. This is computationally intensive, especially when the data dimension $d$ is large.

**Excessive memory usages.** The memory usage of ratio matching is another limitation that cannot be ignored, especially when we learn the energy function using modern GPUs with limited memory. As shown in Eq. (3), the objective function consists of $d$ terms for each training sample. When we do backpropagation, computing the gradient of the objective function *w.r.t.* the learnable parameters of the energy function is required. Therefore, in order to compute such gradient, we have to store the whole computational graph and the intermediate tensors for all of the $d$ terms, thereby leading to excessive memory usages especially if the data dimension $d$ is large.

### 3.2 RATIO MATCHING WITH GRADIENT-GUIDED IMPORTANCE SAMPLING

The key idea is to use the well-known importance sampling technique to reduce the variance of estimating $\mathcal{J}_{RM}(\boldsymbol{\theta})$ with fewer than $d$ terms. The most critical and challenging part of using the importance sampling technique is choosing a good proposal distribution. In this work, we propose to use the gradient of the energy function *w.r.t.* the discrete input $\boldsymbol{x}$ to approximately construct the optimal proposal distribution for importance sampling. We describe the details of our method below.

The objective for each sample $\boldsymbol{x}$, defined by Eq. (3), can be reformulated as

$$\mathcal{J}_{RM}(\boldsymbol{\theta}, \boldsymbol{x}) = d \sum_{i=1}^{d} \frac{1}{d} \left[ e^{E_{\boldsymbol{\theta}}(\boldsymbol{x}) - E_{\boldsymbol{\theta}}(\boldsymbol{x}_{-i})} \right]^2 = d \mathbb{E}_{\boldsymbol{x}_{-i} \sim m(\boldsymbol{x}_{-i})} \left[ e^{E_{\boldsymbol{\theta}}(\boldsymbol{x}) - E_{\boldsymbol{\theta}}(\boldsymbol{x}_{-i})} \right]^2, \qquad (4)$$

where $m(\boldsymbol{x}_{-i}) = \frac{1}{d}$ for $i = 1, \cdots, d$ is a discrete distribution. Thus, the objective of ratio matching for each sample $\boldsymbol{x}$ can be viewed as the expectation of $\left[ e^{E_{\boldsymbol{\theta}}(\boldsymbol{x}) - E_{\boldsymbol{\theta}}(\boldsymbol{x}_{-i})} \right]^2$ over the discrete distribution $m(\boldsymbol{x}_{-i})$. In the original ratio matching method, as described in Section 2.2, we compute such expectation exactly by considering all possible $\boldsymbol{x}_{-i}$, leading to expensive computations and excessive memory usages. Naturally, we can estimate the desired expectation with Monte Carlo method by considering fewer terms sampled based on $m(\boldsymbol{x}_{-i})$. However, such estimation usually has a high variance, and is empirically verified to be ineffective by our experiments in Section 5.

Further, we can apply the importance sampling method to reduce the variance of Monte Carlo estimation. Intuitively, certain values have more impact on the expectation than others. Hence, the estimator variance can be reduced if such important values are sampled more frequently than others. To be specific, instead of sampling based on the distribution $m(\boldsymbol{x}_{-i})$, importance sampling aims to sample from another distribution $n(\boldsymbol{x}_{-i})$, namely, proposal distribution. Formally,

$$\mathcal{J}_{RM}(\boldsymbol{\theta}, \boldsymbol{x}) = d \mathbb{E}_{\boldsymbol{x}_{-i} \sim m(\boldsymbol{x}_{-i})} \left[ e^{E_{\boldsymbol{\theta}}(\boldsymbol{x}) - E_{\boldsymbol{\theta}}(\boldsymbol{x}_{-i})} \right]^2 = d \mathbb{E}_{\boldsymbol{x}_{-i} \sim n(\boldsymbol{x}_{-i})} \frac{m(\boldsymbol{x}_{-i}) \left[ e^{E_{\boldsymbol{\theta}}(\boldsymbol{x}) - E_{\boldsymbol{\theta}}(\boldsymbol{x}_{-i})} \right]^2}{n(\boldsymbol{x}_{-i})}.$$

$$(5)$$

A short derivation of Eq. (5) is in Appendix A. Afterwards, we can apply Monte Carlo estimation based on the proposal distribution $n(\boldsymbol{x}_{-i})$. Specifically, we sample $s$ terms, denoted as $\boldsymbol{x}_{-i}^{(1)}, \cdots, \boldsymbol{x}_{-i}^{(s)}$, according to the proposal distribution $n(\boldsymbol{x}_{-i})$. Note that $s$ is usually chosen to be much smaller than $d$. Then the estimation for $\mathcal{J}_{RM}(\boldsymbol{\theta}, \boldsymbol{x})$ is computed based on these $s$ terms. Formally,

$$\widehat{\mathcal{J}_{RM}(\boldsymbol{\theta}, \boldsymbol{x})}_n = d \frac{1}{s} \sum_{t=1}^{s} \frac{m(\boldsymbol{x}_{-i}^{(t)}) \left[ e^{E_{\boldsymbol{\theta}}(\boldsymbol{x}) - E_{\boldsymbol{\theta}}(\boldsymbol{x}_{-i}^{(t)})} \right]^2}{n(\boldsymbol{x}_{-i}^{(t)})}, \quad \boldsymbol{x}_{-i}^{(t)} \sim n(\boldsymbol{x}_{-i}). \qquad (6)$$

To make it clear, we stop the gradient flowing through the importance weights during backpropagation. It is known that the estimator obtained by Monte Carlo estimation with importance sampling is an unbiased estimator, as the conventional Monte Carlo estimator. The key point of importance sampling is to choose an appropriate proposal distribution $n(\boldsymbol{x}_{-i})$, which determines the variance of the corresponding estimator. Based on Robert et al. (1999), the optimal proposal distribution $n^*(\boldsymbol{x}_{-i})$, which yields the minimum variance, is given by the following fact.

**Fact 1.** *Let* $n^*(\boldsymbol{x}_{-i}) = \frac{\left[ e^{E_{\boldsymbol{\theta}}(\boldsymbol{x}) - E_{\boldsymbol{\theta}}(\boldsymbol{x}_{-i})} \right]^2}{\sum_{k=1}^{d} \left[ e^{E_{\boldsymbol{\theta}}(\boldsymbol{x}) - E_{\boldsymbol{\theta}}(\boldsymbol{x}_{-k})} \right]^2}$ *be a discrete distribution on* $\boldsymbol{x}_{-i}$*, where* $i = 1, \cdots, d$*. Then for any discrete distribution* $n(\boldsymbol{x}_{-i})$ *on* $\boldsymbol{x}_{-i}$*, where* $i = 1, \cdots, d$*, we have* $Var\left( \widehat{\mathcal{J}_{RM}(\boldsymbol{\theta}, \boldsymbol{x})}_{n^*} \right) \leq Var\left( \widehat{\mathcal{J}_{RM}(\boldsymbol{\theta}, \boldsymbol{x})}_n \right)$.

*Proof.* The proof is included in Appendix B. □

To construct the exact optimal proposal distribution $n^*(\boldsymbol{x}_{-i})$, we still have to evaluate the energies of all $\boldsymbol{x}_{-i}$, where $i = 1, \cdots, d$. To avoid such complexity, we propose to leverage the gradient of the energy function *w.r.t.* the discrete input $\boldsymbol{x}$ to approximately construct the optimal proposal distribution. Only $\mathcal{O}(1)$ evaluations of the energy function is needed to construct the proposal distribution.

It is observed by Grathwohl et al. (2021) that many discrete distributions are implemented as continuous and differentiable functions, although they are evaluated only in discrete domains. Grathwohl et al. (2021) further proposes a scalable sampling method for discrete distributions by using the gradients of the underlying continuous functions *w.r.t.* the discrete input. In this study, we extend this idea to improve ratio matching. More specifically, in our case, even though our input $\boldsymbol{x}$ is discrete, our parameterized energy function $E_{\boldsymbol{\theta}}(\cdot)$, such as a neural network, is usually continuous and differentiable. Hence, we can use such gradient information to efficiently and approximately construct the optimal proposal distribution given by Fact 1.

---

**Algorithm 1** Ratio Matching with Gradient-Guided Importance Sampling (RMwGGIS)

---

1: **Input:** Observed dataset $\mathcal{D} = \left\{ \boldsymbol{x}^{(m)} \right\}_{m=1}^{|\mathcal{D}|}$, parameterized energy function $E_{\boldsymbol{\theta}}(\cdot)$, number of samples $s$ for Monte Carlo estimation with importance sampling
2: **for** $\boldsymbol{x} \sim \mathcal{D}$ **do**                              $\triangleright$ Batch training is applied in practice
3:     Compute $E_{\boldsymbol{\theta}}(\boldsymbol{x})$
4:     Compute $\nabla_{\boldsymbol{x}} E_{\boldsymbol{\theta}}(\boldsymbol{x})$
5:     Compute the proposal distribution $\widetilde{n}^*(\boldsymbol{x}_{-i})$                              $\triangleright$ Eq. (10)
6:     Sample $s$ terms, denoted as $\boldsymbol{x}_{-i}^{(1)}, \cdots, \boldsymbol{x}_{-i}^{(s)}$, according to $\widetilde{n}^*(\boldsymbol{x}_{-i})$
7:     Compute $\widehat{\mathcal{J}_{RM}(\boldsymbol{\theta}, \boldsymbol{x})}_{\widetilde{n}^*}$                              $\triangleright$ Eq. (6) (or Eq. (11))
8:     Update $\boldsymbol{\theta}$ based on $\nabla_{\boldsymbol{\theta}} \widehat{\mathcal{J}_{RM}(\boldsymbol{\theta}, \boldsymbol{x})}_{\widetilde{n}^*}$
9: **end for**

---

The basic idea is that we can approximate $E_{\boldsymbol{\theta}}(\boldsymbol{x}_{-i})$ based on the Taylor series of $E_{\boldsymbol{\theta}}(\cdot)$ at $\boldsymbol{x}$, given that $\boldsymbol{x}_{-i}$ is close to $\boldsymbol{x}$ in the data space because they only have differences in one dimension[1]. Formally,

$$E_{\boldsymbol{\theta}}(\boldsymbol{x}_{-i}) \approx E_{\boldsymbol{\theta}}(\boldsymbol{x}) + (\boldsymbol{x}_{-i} - \boldsymbol{x})^T \nabla_{\boldsymbol{x}} E_{\boldsymbol{\theta}}(\boldsymbol{x}). \tag{7}$$

Thus, we can approximately obtain the desired term $E_{\boldsymbol{\theta}}(\boldsymbol{x}) - E_{\boldsymbol{\theta}}(\boldsymbol{x}_{-i})$ in Fact 1 using Eq. (7). Note that $\nabla_{\boldsymbol{x}} E_{\boldsymbol{\theta}}(\boldsymbol{x}) \in \mathbb{R}^d$ contains the information for approximating all $E_{\boldsymbol{\theta}}(\boldsymbol{x}) - E_{\boldsymbol{\theta}}(\boldsymbol{x}_{-i})$, where $i = 1, \cdots, d$. Hence, we can consider the following $d$-dimensional vector

$$(2\boldsymbol{x} - 1) \odot \nabla_{\boldsymbol{x}} E_{\boldsymbol{\theta}}(\boldsymbol{x}) \quad \in \mathbb{R}^d, \tag{8}$$

where $\odot$ denotes the element-wise multiplication. Note that we have $x_i - \bar{x}_i = -1$ if $x_i = 0$ and $x_i - \bar{x}_i = 1$ if $x_i = 1$, which can be unified as $x_i - \bar{x}_i = 2x_i - 1$. Therefore, we have

$$E_{\boldsymbol{\theta}}(\boldsymbol{x}) - E_{\boldsymbol{\theta}}(\boldsymbol{x}_{-i}) \approx [(2\boldsymbol{x} - 1) \odot \nabla_{\boldsymbol{x}} E_{\boldsymbol{\theta}}(\boldsymbol{x})]_i, i = 1, \cdots, d. \tag{9}$$

Afterwards, we can provide a proposal distribution $\widetilde{n}^*(\boldsymbol{x}_{-i})$ as an approximation of the optimal proposal distribution $n^*(\boldsymbol{x}_{-i})$ given by Fact 1. Formally,

$$\widetilde{n}^*(\boldsymbol{x}_{-i}) = \frac{\left[e^{2(2\boldsymbol{x}-1) \odot \nabla_{\boldsymbol{x}} E_{\boldsymbol{\theta}}(\boldsymbol{x})}\right]_i}{\sum_{k=1}^{d} \left[e^{2(2\boldsymbol{x}-1) \odot \nabla_{\boldsymbol{x}} E_{\boldsymbol{\theta}}(\boldsymbol{x})}\right]_k}, \quad i = 1, \cdots, d. \tag{10}$$

Then $\widetilde{n}^*(\boldsymbol{x}_{-i})$ is used as the proposal distribution for Monte Carlo estimation with importance sampling, as in Eq. (6). The overall process of our RMwGGIS method is summarized in Algorithm 1.

### 3.3 COMPARISON BETWEEN RATIO MATCHING AND RMwGGIS

**Time and memory.** Since only $s$ $(s < d)$ terms are considered in the objective function of our RMwGGIS, as shown in Eq. (6), we have better computational efficiency and less memory requirement compared to the original ratio matching method. To be specific, our RMwGGIS only needs $\mathcal{O}(s)$ evaluations of the energy function compared with $\mathcal{O}(d)$ in ratio matching, leading to a linear speedup, which is significant especially when the data is high-dimensional. The improvement in terms of memory usage is similar. In Section 5.1, we compare the real running time and memory usage between ratio matching and our proposed RMwGGIS on datasets with different data dimensions.

**Better optimization?** In Section 3.2, we propose our RMwGGIS based on the motivation to approximate the objective of ratio matching with fewer terms. Although our RMwGGIS can approximate the original ratio matching objective numerically, our objective only includes $s$ terms compared to $d$ terms in the original ratio matching objective. In other words, the objective of ratio matching, as shown in Eq. (3), intuitively pushes up the energies of all $\boldsymbol{x}_{-i}$ for $i = 1, \cdots, d$, while our RMwGGIS only considers pushing up energies of $s$ terms among them, as formulated by Eq. (6). Thus, one may wonder *why our objective is effective for learning EBMs without pushing up the energies of all $d$ terms?* In practice, we even observe that RMwGGIS achieves better density modeling performance

---

[1]We have this assumption because data space is usually high-dimensional. If the number of data dimension is small, we can simply use the original ratio matching method since time and memory are affordable in this case.

than ratio matching. We conjecture that this is because our RMwGGIS can lead to better optimization of Eq. (3) in practice for the following two properties, which are empirically verified in Section 5.

**(1) RMwGGIS introduces stochasticity.** Without involving all $d$ terms in the objective function, our method can introduce stochasticity, which could lead to better optimization in practice. This has the same philosophy as the comparison between mini-batch gradient descent and vanilla gradient descent. The gradient is obtained based on each batch in mini-batch gradient descent, while it is computed over the entire dataset in vanilla gradient descent. It is known that the mini-batch gradient descent usually performs better in practice since the stochasticity introduced by mini-batch training could help escape from the saddle points in non-convex optimization (Ge et al., 2015). Therefore, the stochasticity introduced by sampling only $s$ terms in RMwGGIS might help the optimization especially when $d$ is large.

**(2) RMwGGIS focuses on neighbors with low energies.** Even though only energies of $s$ terms are pushed up in our method, these $s$ terms correspond to the neighboring points that have low energies. According to $n^*(\boldsymbol{x}_{-i})$ given by Fact 1, a neighbor of $\boldsymbol{x}$, denoted as $\boldsymbol{x}_{-i}$, is more likely to be sampled if its corresponding energy value is lower [2]. Hence, we choose to push up the energies of $s$ neighbors according to their current energies. The lower the energy, the more likely it is to be selected. This is intuitively sound because the terms that have low energies are the most offending terms, which should have the higher priorities to be pushed up. In other words, neighbors with lower energies contribute more to the objective function according to Eq. (3). That is, the loss values for these neighbors are larger. Thus, RMwGGIS has the same philosophy as hard negative mining, which pays more attention to hard negative samples during training. More detailed explanation about this connection is provided in Appendix C.

Following the hard negative mining perspective, we observe that the coefficients used in Eq. (6) provide smaller weights for terms with lower energies. In other words, among its selected offending terms (i.e., hard negative samples), it pay least attention to the most offending terms, which is intuitively less effective. Therefore, we further propose the following advanced version as an objective function by removing the coefficients in Eq. (6). Formally,

$$\widehat{\mathcal{J}_{RM}(\boldsymbol{\theta}, \boldsymbol{x})}_{\widetilde{n}^*}^{adv} = \sum_{t=1}^{s} \left[ e^{E_{\boldsymbol{\theta}}(\boldsymbol{x}) - E_{\boldsymbol{\theta}}(\boldsymbol{x}_{-i}^{(t)})} \right]^2, \quad \boldsymbol{x}_{-i}^{(t)} \sim \widetilde{n}^*(\boldsymbol{x}_{-i}). \tag{11}$$

This advanced version is essentially a heuristic extension of the basic version in Eq. (6). It is demonstrated to be more effective in practice. The explanation about this is discussed in Appendix D.

## 4 RELATED WORKS

Learning EBMs has been drawing increasing attention recently. Maximum likelihood training with MCMC sampling, also known as contrastive divergence (Hinton, 2002), is the most representative method. It contrasts samples from training set and samples from the model distribution. To draw samples from the model distribution, we can employ MCMC sampling approaches, such as Langevin dynamics (Welling & Teh, 2011) and Hamiltonian dynamics (Neal et al., 2011). Such methods are further improved and shown to be effective by recent studies (Xie et al., 2016; Gao et al., 2018; Du & Mordatch, 2019; Nijkamp et al., 2019; Grathwohl et al., 2019; Jacob et al., 2020; Qiu et al., 2019; Du et al., 2020a). These methods, however, require the gradient *w.r.t.* the data space to update samples in each MCMC step. Thus, they cannot be applied to discrete data directly. To enable maximum likelihood training with MCMC sampling on discrete data, we can naturally use discrete sampling methods, such as Gibbs sampling and Metropolis-Hastings algorithm (Zanella, 2020), to replace the above gradient-based sampling algorithms. Unfortunately, sampling from a discrete distribution is extremely time-consuming and not scalable. Recently, Dai et al. (2020) develops a learnable sampler parameterized as a local discrete search algorithm to propose negative samples for contrasting. Grathwohl et al. (2021) proposes a scalable sampling method for discrete distributions by surprisingly using the gradient *w.r.t.* the data space, which inspires our work a lot.

An alternative method for learning EBMs is score matching (Hyvärinen & Dayan, 2005; Vincent, 2011; Song et al., 2020; Song & Ermon, 2019), where the scores, *i.e.*, the gradients of the logarithmic

---

[2]Although this only strictly holds for $n^*(\boldsymbol{x}_{-i})$, this can serve as a relaxed explanation for $\widetilde{n}^*(\boldsymbol{x}_{-i})$ as well since $\widetilde{n}^*(\boldsymbol{x}_{-i})$ is a good approximation of $n^*(\boldsymbol{x}_{-i})$. More detailed analysis is included in Appendix C

Table 1: Results on 32-dimensional synthetic discrete data in terms of MMD. The lower the better. The top two results on each dataset are highlighted as **1st** and **2nd**.

| Method | 2spirals | 8gaussians | circles | moons | pinwheel | swissroll | checkerboard |
|---|---|---|---|---|---|---|---|
| Ratio Matching | 0.01514 | 0.10270 | 0.11856 | **0.02901** | 0.31353 | 0.05820 | 0.00059 |
| **RMwGGIS (basic)** | **0.01099** | **0.09763** | **0.11017** | 0.03111 | **0.27885** | **0.05176** | **0.00050** |
| **RMwGGIS (advanced)** | **0.00876** | **0.08414** | **0.10230** | **0.02787** | **0.26188** | **0.04477** | **0.00026** |

probability distribution *w.r.t.* the data space, of the energy function are forced to match the scores of the training data. Ratio matching (Hyvärinen, 2007; Lyu, 2009) is obtained by extending the idea of score matching to discrete data. Our work is motivated by the limitations of ratio matching, as analyzed in Section 3.1. Stochastic ratio matching (Dauphin & Bengio, 2013) also aims to make ratio matching more efficient by considering the sparsity of input data. Stochastic ratio matching is limited to sparse data, while our method is effective for general EBMs

There are some other methods for learning EBMs, such as noise contrastive estimation (Gutmann & Hyvärinen, 2010; Bose et al., 2018; Ceylan & Gutmann, 2018; Gao et al., 2020) and learning the stein discrepancy (Grathwohl et al., 2020). We recommend readers to refer to Song & Kingma (2021) for a comprehensive introduction on learning EBMs. We note that several works (Elvira et al., 2015; Schuster, 2015) use the gradient information of the target distribution to iteratively optimize the proposal distributions for adaptive importance sampling. However, compared to our method, they can only applied to continuous distributions and require expensive iterative processes.

## 5 EXPERIMENTS

### 5.1 DENSITY MODELING ON SYNTHETIC DISCRETE DATA

**Setup.** For both quantitative results and qualitative visualization, we follow the experimental setting of Dai et al. (2020) for density modeling on synthetic discrete data. We firstly draw 2D data points from 2D continuous space according to some unknown distribution $\hat{p}$, which can be naturally visualized. Then, we convert each 2D data point $\hat{x} \in \mathbb{R}^2$ to a discrete data point $x \in \{0,1\}^d$, where $d$ is the desired number of data dimensions. To be specific, we transform each dimension of $\hat{x}$, which is a floating-point number, into a $\frac{d}{2}$-bit Gray code[3] and concatenate the results to obtain a $d$-bit vector $x$. Thus, the unknown distribution in discrete space is $p(x) = \hat{p}\left(\left[\text{GrayToFloat}(x_{1:\frac{d}{2}}), \text{GrayToFloat}(x_{\frac{d}{2}+1:d})\right]\right)$. This density modeling task is challenging since the transformation from $\hat{x}$ to $x$ is non-linear.

To quantitatively evaluate the performance of density modeling, we adopt the maximum mean discrepancy (MMD) (Gretton et al., 2012) with a linear kernel corresponding to *(d-HammingDistance)* to compare distributions. In our case, particularly, the MMD metric is computed based on $4000$ samples, drawn from the learned energy function via Gibbs sampling, and the same number of samples from the training set. Lower MMD indicates that the distribution defined by the learned energy function is closer to the unknown data distribution. In addition, in order to qualitatively visualize the learned energy function, we firstly uniformly obtain $10k$ data points from 2D continuous space. Afterwards, they are converted into bit vectors and evaluated by the learned energy function. Subsequently, we can visualize the obtained corresponding energies in 2D space.

The energy function is parameterized by an MLP with the Swish (Ramachandran et al., 2017) activation and 256 hidden dimensions. The number of samples $s$, involved in the objective functions of our RMwGGIS method, is set to be 10. In the following, we compare our basic and advanced RMwGGIS methods, as formulated in Eq. (6) and Eq. (11) respectively, with the original ratio matching method (Hyvärinen, 2007; Lyu, 2009).

**Quantitative and qualitative results.** The quantitative results on 32-dimensional datasets are shown in Table 1. Our RMwGGIS, including the basic and the advanced versions, consistently outperforms the original ratio matching by large margins, which demonstrates that it is effective for our proposed gradient-guided importance sampling to stochastically push up neighbors with low energies. This

---

[3] https://en.wikipedia.org/wiki/Gray_code

verifies our analysis in Section 3.3. In Figure 1, we qualitatively visualize the learned energy functions of our proposed RMwGGIS. It is observed that EBMs learned by our method can fit the data distribution accurately. Note that we choose $d = 32$ for quantitative evaluation because Gibbs sampling cannot obtain appropriate samples from the learned energy function with an affordable time budget if the data dimension is too high, thus leading to invalid MMD results. We will compare the results on higher-dimensional data in the following by observing the qualitative visualization. To further demonstrate that the performance improvement of RMwGGIS over ratio matching is brought by better optimization, we show that energy functions learned with our methods actually lead to lower value for the objective function defined by Eq. (3). The details are included in Appendix E.

**Observations on higher-dimensional data.** As analyzed in Section 3.3, the advantages of our approach can be greater on higher-dimensional data. To evaluate this, we conduct experiments on the 256-dimensional *2spirals* dataset, and visualize the learned energy functions corresponding to different learning iterations. We construct a ablation method, named as RMwRAND, which estimates the original ratio matching objective by randomly sampling $s = 10$ terms.

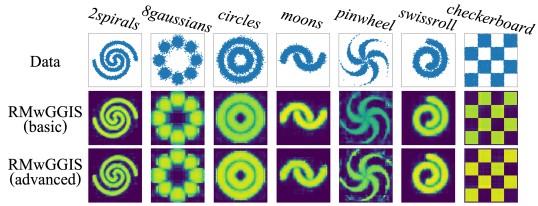

Figure 1: Visualization of learned energy functions on 32-dimensional synthetic discrete datasets.

The only difference between our RMwGGIS method and RMwRAND is that we focus more on the terms corresponding to low energies, based on our proposed gradient-guided importance sampling.

As shown in Figure 4, Appendix F, our RMwGGIS accurately captures the data distribution, while the original ratio matching method cannot. This further verifies that RMwGGIS leads to better optimization than ratio matching especially when the data dimension is high, as analyzed in Section 3.3. In addition, although RMwRAND can also introduce stochasticity as RMwGGIS by randomly sampling, it fails to capture the data distribution. This observation is intuitively reasonable since randomly pushing up $s = 10$ terms among $d = 256$ terms leads to large variance. Instead, our RMwGGIS performs well since we focus on pushing up terms with low energies, which are the most offending terms and should be pushed up first. Overall, these experiments can show the superiority of RMwGGIS endowed by our proposed gradient-guided importance sampling on high-dimensional data.

**Selection of** $s$. Note that we did not perform extensive tuning for $s$, although tuning it might bring performance improvement. To further show the influence of $s$ and the effectiveness of RMwGGIS (advanced) over RMwRAND. We perform an experiment to train both models on 256-dimensional *2spirals* with $s = 5, 10, 50, 100$. The visualization of the learned energy functions is shown in Figure 2. Our RMwGGIS is observed to achieve good density modeling performance that is robust to the selection of $s$. In contrast, RMwRAND fails

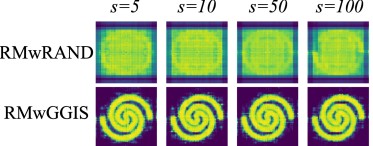

Figure 2: Comparison of RMwG-GIS(advanced) and RMwRAND with several configurations of $s$.

to capture the data distribution even with $s = 100$. This further demonstrates the effectiveness and robustness of our proposed gradient-guided importance sampling strategy.

**Running time and memory usage.** To verify RMwGGIS has better efficiency than ratio matching, we compare the real running time and memory usage on datasets of various dimensions. Specifically, we construct several *2spirals* datasets with different data dimensions and train parameterized energy functions using ratio matching and our RMwGGIS, respectively. We choose batch size to be 256. The reported time corresponds to the average training time per batch. For RMwGGIS, we report the results of the advanced version.

As summarized in Table 5, Appendix G, our RMwGGIS is much more efficient in terms of running time and memory usage, compared to ratio matching. In addition, the improvement is more obvious with the increasing of data dimension. Specifically, compared with ratio matching, our RMwGGIS can achieve 6.2 times speedup and save 90.1% memory usage on the 2048-dimensional dataset.

## 5.2 GRAPH GENERATION

**Setup.** We further evaluate our RMwGGIS on graph generation using the *Ego-small* dataset (You et al., 2018). It is a set of one-hop ego graphs, where the number of nodes $4 \leq |V| \leq 18$, obtained

from the Citeseer network (Sen et al., 2008). Following the experimental setting of You et al. (2018) and Liu et al. (2019), $80\%$ of the graphs are used for training and the rest for testing. New graphs can be generated via Gibbs sampling on the learned energy function. To evaluate the graph generation performance based on the generated graphs and the test graphs, we calculate MMD over three statistics, *i.e.*, degrees, clustering coefficients, and orbit counts, as proposed in You et al. (2018).

We parameterize the energy function by a 5-layer R-GCN (Schlichtkrull et al., 2018) model with the Swish activation and 32 hidden dimensions, whose input is the upper triangle of the graph adjacency matrix. The number of samples $s$ used in our RMwGGIS objective is $50$. We apply our advanced version to learn the energy function since it is more effective in Section 5.1. Besides ratio matching, we consider the recent proposed method EBM (GWG) (Grathwohl et al., 2021) as a baseline. We also consider the recent works developed for graph generation as baselines, including GraphVAE (Simonovsky & Komodakis, 2018), DeepGMG (Li et al., 2018), GraphRNN (You et al., 2018),

Table 2: Graph generation results in terms of MMD. *Avg.* denotes the average over three MMD results.

| Method | Degree | Cluster | Orbit | Avg. |
|---|---|---|---|---|
| GraphVAE | 0.130 | 0.170 | 0.050 | 0.117 |
| DeepGMG | 0.040 | 0.100 | 0.020 | 0.053 |
| GraphRNN | 0.090 | 0.220 | 0.003 | 0.104 |
| GNF | 0.030 | 0.100 | 0.001 | 0.044 |
| EDP-GNN | 0.052 | 0.093 | 0.007 | 0.050 |
| GraphAF | 0.030 | 0.110 | 0.001 | 0.047 |
| GraphDF | 0.040 | 0.130 | 0.010 | 0.060 |
| EBM (GWG) | 0.093 | 0.027 | 0.053 | 0.058 |
| Ratio Matching | 0.062 | 0.066 | 0.008 | 0.045 |
| **RMwGGIS** | 0.044 | 0.059 | 0.013 | **0.039** |

GNF (Liu et al., 2019), EDP-GNN (Niu et al., 2020), GraphAF (Shi et al., 2019), and GraphDF (Luo et al., 2021). The detailed setup is provided in Appendix H

**Quantitative and qualitative results.** As summarized in Table 2, our RMwGGIS outperforms baselines in terms of the average over three MMD results. This shows that our method can learn EBMs to generate graphs that align with various characteristics of the training graphs. The generated samples are visualized in Figure 5, Appendix I. It can be observed that the generated samples are realistic one-hop ego graphs that have similar characteristics as the training samples.

## 5.3 TRAINING ISING MODELS

To further demonstrate the scaling ability of our method and compare with recent baselines more thoroughly, we use our

Table 3: Comparison of training Ising models.

| MCMC #Steps | | 5 | 10 | 25 | 50 | 100 | Ratio Matching | **RMwGGIS** |
|---|---|---|---|---|---|---|---|---|
| log(RMSE) | EBM (Gibbs) | −1.60 | −1.90 | −2.50 | −3.00 | −3.60 | −3.99 | −4.06 |
| | EBM (GWG) | −4.02 | −4.49 | −4.87 | −4.94 | −5.05 | | |
| Time/iter | EBM (Gibbs) | 263.5ms | 437.3ms | 1113.2ms | 2524.9ms | 4670.1ms | 450.7ms | **13.9**ms |
| | EBM (GWG) | 37.5ms | 63.4ms | 100.5ms | 222.6ms | 395.7ms | | |

RMwGGIS to train the Ising model with a 2D cyclic lattice structure, following Grathwohl et al. (2021). We consider a $25 \times 25$ lattice, thus leading to a 625-dimension problem, which can be used to evaluate the ability of scaling to high-dimensional problems. We compare methods in terms of the RMSE between the inferred connectivity matrix $\hat{J}$ and the true $J$ and the running time per iteration. The experimental details are included in Appendix J. As shown in Table 3, our RMwGGIS is more effective than ratio matching and EBM (Gibbs) with various sample steps. The recently proposed EBM (GWG) (Grathwohl et al., 2021) achieves better RMSE than ours. In terms of running time, our method is much more efficient than baselines since we avoid the expensive MCMC sampling during training and do not have to consider flipping all dimensions as ratio matching. According to this experiment, one future direction could be further improving the effectiveness of RMwGGIS while preserving the efficiency advantage.

## 6 CONCLUSION

We propose ratio matching with gradient-guided importance sampling (RMwGGIS) for learning EBMs on binary discrete data. In particular, we use the gradient of the energy function *w.r.t.* the discrete input space to guide the importance sampling for estimating the original ratio matching objective. We further connect our method with hard negative mining, and obtain the advanced version of RMwGGIS. Compared to ratio matching, our RMwGGIS methods, including the basic and advanced versions, are more efficient in terms of computation and memory usage, and are shown to be more effective for density modeling. Extensive experiments on synthetic data density modeling, graph generation, and Ising model training demonstrate that our RMwGGIS achieves significant improvements over previous methods in terms of both effectiveness and efficiency.

## ACKNOWLEDGMENTS

This work was supported in part by National Science Foundation grant IIS-1908220.

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

## A  THE DETAILED DERIVATION OF EQ. (5)

The detailed derivation of Eq. (5) is as follows.

$$
\begin{aligned}
\mathcal{J}_{RM}(\boldsymbol{\theta}, \boldsymbol{x}) &= d\mathbb{E}_{\boldsymbol{x}_{-i} \sim m(\boldsymbol{x}_{-i})} \left[ e^{E_{\boldsymbol{\theta}}(\boldsymbol{x}) - E_{\boldsymbol{\theta}}(\boldsymbol{x}_{-i})} \right]^2 \\
&= d \sum_{i=1}^{d} m(\boldsymbol{x}_{-i}) \left[ e^{E_{\boldsymbol{\theta}}(\boldsymbol{x}) - E_{\boldsymbol{\theta}}(\boldsymbol{x}_{-i})} \right]^2 \\
&= d \sum_{i=1}^{d} \frac{m(\boldsymbol{x}_{-i}) \left[ e^{E_{\boldsymbol{\theta}}(\boldsymbol{x}) - E_{\boldsymbol{\theta}}(\boldsymbol{x}_{-i})} \right]^2}{n(\boldsymbol{x}_{-i})} n(\boldsymbol{x}_{-i}) \\
&= d\mathbb{E}_{\boldsymbol{x}_{-i} \sim n(\boldsymbol{x}_{-i})} \frac{m(\boldsymbol{x}_{-i}) \left[ e^{E_{\boldsymbol{\theta}}(\boldsymbol{x}) - E_{\boldsymbol{\theta}}(\boldsymbol{x}_{-i})} \right]^2}{n(\boldsymbol{x}_{-i})}.
\end{aligned}
\tag{12}
$$

## B  PROOF OF FACT 1

*Proof.* According to Eq. (6), we have

$$
Var\left( \widehat{\mathcal{J}_{RM}(\boldsymbol{\theta}, \boldsymbol{x})}_n \right) = \frac{d^2}{s} Var \left( \frac{m(\boldsymbol{x}_{-i}) \left[ e^{E_{\boldsymbol{\theta}}(\boldsymbol{x}) - E_{\boldsymbol{\theta}}(\boldsymbol{x}_{-i})} \right]^2}{n(\boldsymbol{x}_{-i})} \right).
\tag{13}
$$

Then we can compare the variance of the estimator based on $n^*(\boldsymbol{x}_{-i})$ and $n(\boldsymbol{x}_{-i})$. Formally,

$$
Var\left( \widehat{\mathcal{J}_{RM}(\boldsymbol{\theta}, \boldsymbol{x})}_{n^*} \right)
\tag{14}
$$

$$
= \frac{d^2}{s} Var \left( \frac{m(\boldsymbol{x}_{-i}) \left[ e^{E_{\boldsymbol{\theta}}(\boldsymbol{x}) - E_{\boldsymbol{\theta}}(\boldsymbol{x}_{-i})} \right]^2}{n^*(\boldsymbol{x}_{-i})} \right)
\tag{15}
$$

$$
= \frac{d^2}{s} \left\{ \mathbb{E}_{\boldsymbol{x}_{-i} \sim n^*(\boldsymbol{x}_{-i})} \left[ \frac{m(\boldsymbol{x}_{-i}) \left[ e^{E_{\boldsymbol{\theta}}(\boldsymbol{x}) - E_{\boldsymbol{\theta}}(\boldsymbol{x}_{-i})} \right]^2}{n^*(\boldsymbol{x}_{-i})} \right]^2 - \left[ \mathbb{E}_{\boldsymbol{x}_{-i} \sim n^*(\boldsymbol{x}_{-i})} \frac{m(\boldsymbol{x}_{-i}) \left[ e^{E_{\boldsymbol{\theta}}(\boldsymbol{x}) - E_{\boldsymbol{\theta}}(\boldsymbol{x}_{-i})} \right]^2}{n^*(\boldsymbol{x}_{-i})} \right]^2 \right\}
\tag{16}
$$

$$
= \frac{d^2}{s} \left\{ \mathbb{E}_{\boldsymbol{x}_{-i} \sim n^*(\boldsymbol{x}_{-i})} \left[ \frac{m(\boldsymbol{x}_{-i}) \left[ e^{E_{\boldsymbol{\theta}}(\boldsymbol{x}) - E_{\boldsymbol{\theta}}(\boldsymbol{x}_{-i})} \right]^2}{n^*(\boldsymbol{x}_{-i})} \right]^2 - \left[ \frac{\mathcal{J}_{RM}(\boldsymbol{\theta}, \boldsymbol{x})}{d} \right]^2 \right\}
\tag{17}
$$

$$
= \frac{d^2}{s} \left\{ \frac{1}{d} \sum_{i=1}^{d} n^*(\boldsymbol{x}_{-i}) \left[ \frac{m(\boldsymbol{x}_{-i}) \left[ e^{E_{\boldsymbol{\theta}}(\boldsymbol{x}) - E_{\boldsymbol{\theta}}(\boldsymbol{x}_{-i})} \right]^2}{n^*(\boldsymbol{x}_{-i})} \right]^2 - \left[ \frac{\mathcal{J}_{RM}(\boldsymbol{\theta}, \boldsymbol{x})}{d} \right]^2 \right\}
\tag{18}
$$

$$
= \frac{d^2}{s} \left\{ \frac{1}{d} \sum_{i=1}^{d} m^2(\boldsymbol{x}_{-i}) \left[ e^{E_{\boldsymbol{\theta}}(\boldsymbol{x}) - E_{\boldsymbol{\theta}}(\boldsymbol{x}_{-i})} \right]^2 \sum_{k=1}^{d} \left[ e^{E_{\boldsymbol{\theta}}(\boldsymbol{x}) - E_{\boldsymbol{\theta}}(\boldsymbol{x}_{-k})} \right]^2 - \left[ \frac{\mathcal{J}_{RM}(\boldsymbol{\theta}, \boldsymbol{x})}{d} \right]^2 \right\}
\tag{19}
$$

$$
= \frac{d^2}{s} \left\{ \frac{1}{d} \left[ \sum_{i=1}^{d} m(\boldsymbol{x}_{-i}) \left[ e^{E_{\boldsymbol{\theta}}(\boldsymbol{x}) - E_{\boldsymbol{\theta}}(\boldsymbol{x}_{-i})} \right]^2 \right]^2 - \left[ \frac{\mathcal{J}_{RM}(\boldsymbol{\theta}, \boldsymbol{x})}{d} \right]^2 \right\}
\tag{20}
$$

$$
= \frac{d^2}{s} \left\{ \frac{1}{d} \left[ \sum_{i=1}^{d} \frac{m(\boldsymbol{x}_{-i}) \left[ e^{E_{\boldsymbol{\theta}}(\boldsymbol{x}) - E_{\boldsymbol{\theta}}(\boldsymbol{x}_{-i})} \right]^2}{n(\boldsymbol{x}_{-i})} \sqrt{n(\boldsymbol{x}_{-i})} \sqrt{n(\boldsymbol{x}_{-i})} \right]^2 - \left[ \frac{\mathcal{J}_{RM}(\boldsymbol{\theta}, \boldsymbol{x})}{d} \right]^2 \right\}
\tag{21}
$$

$$
\leq \frac{d^2}{s} \left\{ \frac{1}{d} \sum_{i=1}^{d} \left[ \frac{m(\boldsymbol{x}_{-i}) \left[ e^{E_{\boldsymbol{\theta}}(\boldsymbol{x}) - E_{\boldsymbol{\theta}}(\boldsymbol{x}_{-i})} \right]^2}{n(\boldsymbol{x}_{-i})} \sqrt{n(\boldsymbol{x}_{-i})} \right]^2 \sum_{i=1}^{d} n(\boldsymbol{x}_{-i}) - \left[ \frac{\mathcal{J}_{RM}(\boldsymbol{\theta}, \boldsymbol{x})}{d} \right]^2 \right\}
\tag{22}
$$

$$= \frac{d^2}{s} \left\{ \frac{1}{d} \sum_{i=1}^{d} n(\boldsymbol{x}_{-i}) \left[ \frac{m(\boldsymbol{x}_{-i}) \left[ e^{E_{\boldsymbol{\theta}}(\boldsymbol{x}) - E_{\boldsymbol{\theta}}(\boldsymbol{x}_{-i})} \right]^2}{n(\boldsymbol{x}_{-i})} \right]^2 - \left[ \frac{\mathcal{J}_{RM}(\boldsymbol{\theta}, \boldsymbol{x})}{d} \right]^2 \right\} \tag{23}$$

$$= \frac{d^2}{s} \left\{ \mathbb{E}_{\boldsymbol{x}_{-i} \sim n(\boldsymbol{x}_{-i})} \left[ \frac{m(\boldsymbol{x}_{-i}) \left[ e^{E_{\boldsymbol{\theta}}(\boldsymbol{x}) - E_{\boldsymbol{\theta}}(\boldsymbol{x}_{-i})} \right]^2}{n(\boldsymbol{x}_{-i})} \right]^2 - \left[ \frac{\mathcal{J}_{RM}(\boldsymbol{\theta}, \boldsymbol{x})}{d} \right]^2 \right\} \tag{24}$$

$$= Var\left( \widehat{\mathcal{J}_{RM}(\boldsymbol{\theta}, \boldsymbol{x})}_n \right). \tag{25}$$

Eq. (17) can be derived because the estimator is unbiased no matter what proposal distribution is applied. Eq. (19) is obtained by choosing $n^*(\boldsymbol{x}_{-i}) = \frac{\left[ e^{E_{\boldsymbol{\theta}}(\boldsymbol{x}) - E_{\boldsymbol{\theta}}(\boldsymbol{x}_{-i})} \right]^2}{\sum_{k=1}^{d} \left[ e^{E_{\boldsymbol{\theta}}(\boldsymbol{x}) - E_{\boldsymbol{\theta}}(\boldsymbol{x}_{-k})} \right]^2}$. Eq. (20) holds since $m(\boldsymbol{x}_{-i}) = \frac{1}{d}$ for $i = 1, \cdots, d$. To derive Eq. (22), we apply the Cauchy-Schwarz inequality $\left( \sum_{i=1}^{d} a_i b_i \right)^2 \leq \left( \sum_{i=1}^{d} a_i^2 \right) \left( \sum_{i=1}^{d} b_i^2 \right)$. This completes the proof of Fact 1.

## C  CONNECTION WITH HARD NEGATIVE MINING

Here, we provide an insight to understand why the second property, described in Section 3.3, can lead to better optimization, by connecting it with hard sample mining.

Hard sample mining (Felzenszwalb et al., 2009; Rowley et al., 1998) has been widely applied to train deep neural networks (Shrivastava et al., 2016). Our RMwGGIS is particularly highly related to hard negative training strategies. The basic idea for hard negative mining is to pay more attention to hard negative samples during training, which can usually achieve better performance since it can reduce false positives. In our setting of discrete EBMs, each training sample $\boldsymbol{x}$ is a positive sample, and its energy should be pushed down. For each positive sample $\boldsymbol{x}$, all $\boldsymbol{x}_{-i}$ for $i = 1, 2, \cdots, d$ are negative samples, and their energy should be pushed up. Our RMwGGIS with the specific proposal distribution shown in Eq. (10) can approximately choose the $\boldsymbol{x}_{-i}$'s that currently have low energies with larger probabilities. This has the same philosophy as hard negative mining. Specifically, in our case, $\boldsymbol{x}_{-i}$'s with low energies are hard negative samples since they are the most offending terms, which are close to the positive sample $\boldsymbol{x}$ and have low energies.

Since our proposal distribution defined in Eq. (10) approximates the provable optimal proposal distribution given in Theorem 1, our proposal distribution thus *approximately* performs "hard negative mining". The natural follow-up question is *how accurate is the approximation and how does it affect the learning process?* We answer this question by analyzing the following two stages during learning, which can intuitively show that our RMwGGIS is technically sound.

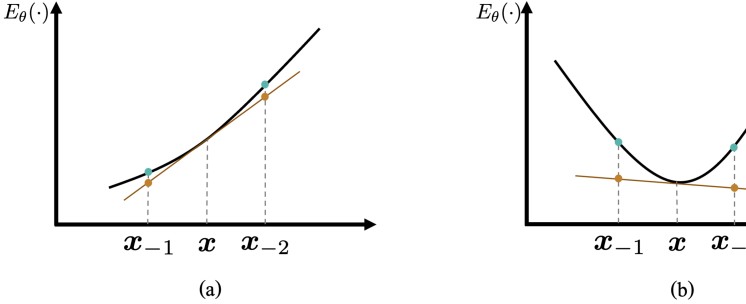

(a)  (b)

Figure 3: An intuitive illustration of the approximation: (a) Stage I and (b) Stage II. The black curves denote the energy functions. Green nodes and brown nodes represent the true energy values and approximated values of neighbors, respectively. Note that the approximated values are obtained based on Taylor series of $E_{\theta}(\cdot)$ at $\boldsymbol{x}$, as shown in Eq. (7). For clarity, we only show two neighbors of $\boldsymbol{x}$ in this figure, but this illustration can also be extended to include all neighbors.

**Stage I.** As shown in Figure 3 (a), at the early stage of learning, the energy function is not learned well, thus the energy $E_{\theta}(\boldsymbol{x})$ of positive sample $\boldsymbol{x}$ is not smaller than its all neighbors. In this case,

Table 4: Comparison of resulting objective values. The top two lowest values on each dataset are highlighted as **1st** and **2nd**.

| Method | 2spirals | 8gaussians | circles | moons | pinwheel | swissroll | checkerboard |
|---|---|---|---|---|---|---|---|
| Ratio Matching | 46.02 | 39.26 | 31.82 | 28.57 | 28.50 | 37.52 | **26.05** |
| **RMwGGIS (basic)** | **26.84** | **30.15** | **29.89** | **27.80** | **28.08** | **32.20** | 26.09 |
| **RMwGGIS (advanced)** | **27.15** | **29.31** | **29.54** | **27.75** | **27.30** | **29.45** | **26.06** |

there are some neighbors of $\boldsymbol{x}$ which have lower energies than $E_\theta(\boldsymbol{x})$, such as $\boldsymbol{x}_{-1}$ in Figure 3 (a). Therefore, "hard negative mining" is in demand in this stage. Under this situation, our approximated energies of neighbors could help to perform "hard negative mining". To be specific, the estimated energies of neighbors are close to the true energies, and the estimated energy of $\boldsymbol{x}_{-1}$ is much lower than the estimated energy of $\boldsymbol{x}_{-2}$. Thus, our proposal distribution will sample $\boldsymbol{x}_{-1}$ with a higher probability. This actually works as "hard negative mining" since $\boldsymbol{x}_{-1}$ is the current most offending term, *i.e.*, the so-called hard negative sample.

**Stage II.** After learning for a while, we can obtain a relatively good energy function, where the positive sample $\boldsymbol{x}$ locates in the low energy area compared to its local neighbors. In this case our approximation is less accurate. Fortunately, in this case, "hard negative mining" is not that necessary since there do not exist many offending terms. Specifically, as shown in Figure 3 (b), the energies of $\boldsymbol{x}_{-1}$ and $\boldsymbol{x}_{-2}$ are safely higher than $E_\theta(\boldsymbol{x})$.

Even though the above analysis is based on a simplified example, we believe it can serve as a good intuitive understanding of why our RMwGGIS performs better than ratio matching.

## D  WHY DOES THE ADVANCED VERSION PERFORM BETTER?

Here, we provide an intuitive explanation on why our advanced version usually performs better than the basic version.

Following our analysis of the connection between our method and hard negative mining, as described in Appendix C, it is obvious that both basic version (*i.e.*, Eq. (6)) and advanced version (*i.e.*, Eq. (11)) perform "hard negative mining". The difference lies in the coefficients for different terms. Specifically, the advanced version gives the same weights to all sampled terms. In contrast, the basic version provides a weight $\frac{m(\boldsymbol{x}_{-i}^{(t)})}{n(\boldsymbol{x}_{-i}^{(t)})}$ to each sampled term $\boldsymbol{x}_{-i}^{(t)}$, as shown in Eq. (6). Note that $m(\boldsymbol{x}_{-i}^{(t)}) = \frac{1}{d}$ for all terms and $n(\boldsymbol{x}_{-i}^{(t)})$ would be larger if $\boldsymbol{x}_{-i}^{(t)}$ has lower energy. Hence, the basic version provides smaller weights for terms with lower energies. In other words, among its selected offending terms (*i.e.*, hard negative samples), it pay least attention to the most offending terms, which could be less effective than the advanced version with equal weights. This could explain why our advanced RMwGGIS usually performs better than the basic version.

## E  COMPARISON OF ACHIEVED OBJECTIVE VALUES

Specifically, for all the learned energy functions in Table 1, we sample 4000 data points on each dataset and evaluate the resulting objective value defined by Eq. (3). The results are summarized in Table 4. We can observe that our basic and advanced RMwGGIS indeed achieve lower objective values, which further demonstrates that our proposed RMwGGIS can lead to better optimization. This is quite natural and straightforward since our method focus on neighbors with low energies. Specifically, according to the objective function of ratio matching (*i.e.*, Eq. (3)), neighbors with lower energies contribute more to the objective function. In other words, the loss values for these neighbors are larger. This explains why our methods, focusing on reducing the loss values (*i.e.*, pushing up energies) of these neighbors, indeed lead to lower value for the objective function defined by Eq. (3).

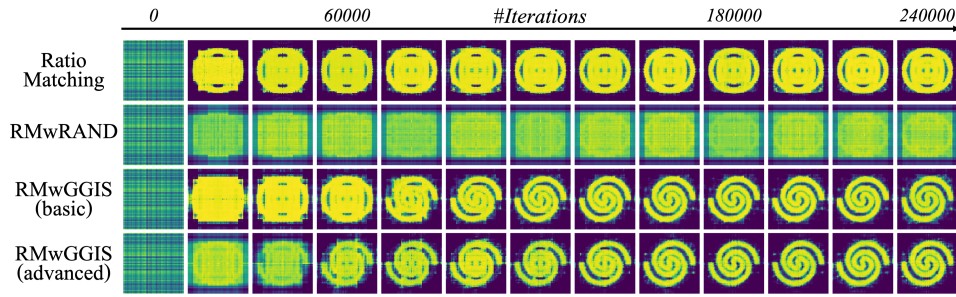

Figure 4: Visualization of learned energy functions *w.r.t.* number of learning iterations on the 256-dimensional *2spirals* dataset.

Table 5: Comparison between ratio matching and our RMwGGIS on *2spirals* datasets with different dimensions in terms of running time and memory usage.

| | # *Data Dimensions* | *32* | *64* | *128* | *256* | *512* | *1024* | *2048* |
|---|---|---|---|---|---|---|---|---|
| Time | Ratio Matching | 63.9ms | 106.5ms | 185.6ms | 372.9ms | 735.2ms | 1390.1ms | 2684.1ms |
| | **RMwGGIS** | **41.2**ms | **47.2**ms | **58.8**ms | **86.9**ms | **137.7**ms | **244.9**ms | **434.1**ms |
| | **Speedup** | **1.6**× | **2.3**× | **3.2**× | **4.3**× | **5.3**× | **5.7**× | **6.2**× |
| Memory | Ratio Matching | 957MB | 1031MB | 1189MB | 1545MB | 2315MB | 4237MB | 9633MB |
| | **RMwGGIS** | **891**MB | **893**MB | **893**MB | **915**MB | **919**MB | **931**MB | **951**MB |
| | **Memory Saving** | **6.9**% | **13.4**% | **24.9**% | **40.8**% | **60.3**% | **78.0**% | **90.1**% |

# F VISUALIZATION OF LEARNED ENERGY FUNCTIONS ON HIGHER-DIMENSIONAL DATA

The learned energy functions *w.r.t.* number of learning iterations on the 256-dimensional *2spirals* dataset are qualitatively visualized in Figure 4

# G COMPARISON OF RUNNING TIME AND MEMORY USAGE

The comparison of running time and memory usage are summarized in Table 5.

# H DETAILED SETTINGS OF GRAPH GENERATION

The results of baselines in Table 2 are reported from You et al. (2018), Liu et al. (2019), Niu et al. (2020), Shi et al. (2019), and Luo et al. (2021). We obtain the result of EBM (GWG) by using their official implementation, and the detailed settings are provided as follows.

We use the official open-sourced implementation[4] of EBM (GWG) to perform its graph generation experiment. We train models with persistent contrastive divergence (Tieleman, 2008) with a buffer size of 200 samples. We use the Adam optimizer (Kingma & Ba, 2015) with a learning rate of $1e$-4 and a batch size of 200. The following hyperparameters are tuned and the finally chosen ones are underlined: buffer initialization rate $\in \{0, 0.2, \underline{0.4}, 0.6\}$ and MCMC steps $\in \{100, 200, \underline{500}, 1000, 2000\}$.

# I VISUALIZATION OF GENERATED GRAPHS

Generated graph samples are shown in Figure 5.

---

[4]https://github.com/wgrathwohl/GWG_release

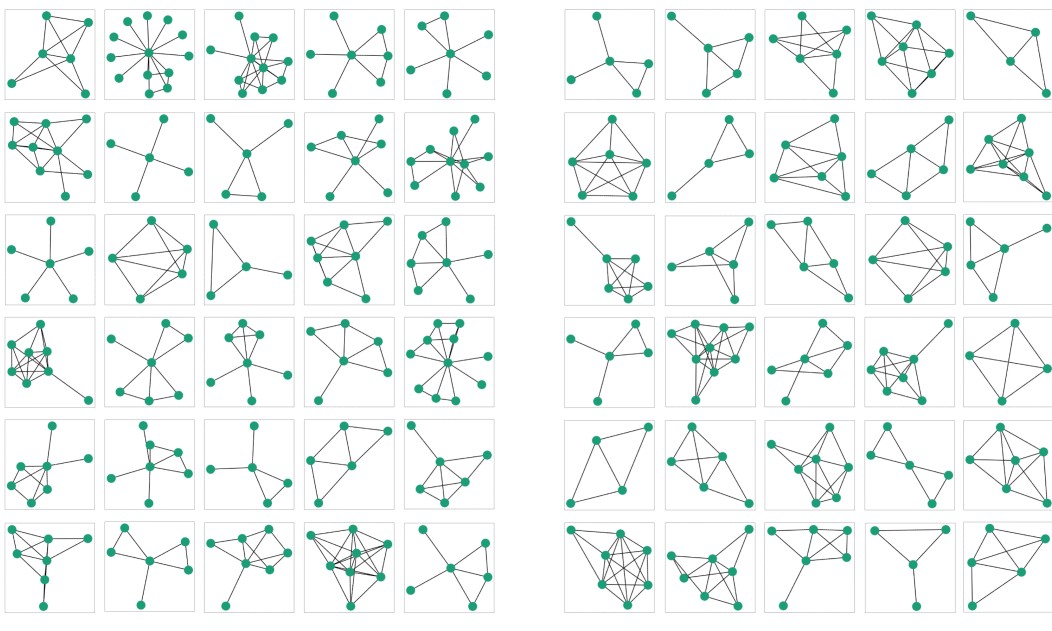

Training data                                      RMwGGIS samples

Figure 5: Visualization of training data and samples drawn from the energy function learned by our RMwGGIS for graph generation.

## J    DETAILS OF TRAINING ISING MODELS

**Lattice Ising Models.** Ising models (Cipra, 1987) is firstly developed to model the spin magnetic particles (Ising, 1925). For Ising models, our energy function can be naturally defined as

$$E(\boldsymbol{x}) = -\boldsymbol{x}^T \boldsymbol{J} \boldsymbol{x} - \boldsymbol{b}^T \boldsymbol{x}, \tag{26}$$

where $\boldsymbol{J}$ and $\boldsymbol{b}$ are the parameters. $\boldsymbol{J}$ is the connectivity matrix which indicates the correlation across dimensions in $\boldsymbol{x}$. We follow one specific setting in Grathwohl et al. (2021), where all of the non-zero entries of $\boldsymbol{J}$ are identical (denoted as $\sigma$) and $\boldsymbol{J}$ is the adjacency matrix of a cyclic 2D lattice structure. Therefore,

$$E(\boldsymbol{x}) = -\sigma \boldsymbol{x}^T \boldsymbol{J} \boldsymbol{x} - \boldsymbol{b}^T \boldsymbol{x}. \tag{27}$$

**Setup.** We follow Grathwohl et al. (2021) for our experimental setting. To be specific, we create a model using a $25 \times 25$ lattice and $\sigma = 0.25$, thus leading to a $625$ dimensional distribution. For training the model, 2000 examples are generated via $1,000,000$ steps of Gibbs sampling. We apply our proposed RMwGGIS method to train the model. The number of samples $s$ used in our RMwGGIS objective is set to $10$. We use Adam optimizer (Kingma & Ba, 2015) with a learning rate of $1e\text{-}4$ and a batch size of $100$. $\ell 1$ penalty with strength $0.01$ is used to encourage sparsity. In terms of baselines, in addition to ratio matching, we further consider the approaches which train discrete EBMs with persistent contrastive divergence (Tieleman, 2008). The number of steps for MCMC per training iteration is $\in \{5, 10, 25, 50, 100\}$. The samplers are Gibbs and Gibbs-With-Gradient (GWG) (Grathwohl et al., 2021). Results of EBM (GWG) are obtained by running the official implementation from Grathwohl et al. (2021). Results of EBM (Gibbs) are obtained by reading from Figure 6 in Grathwohl et al. (2021).

**Evaluation.** We evaluate the performance by computing the root-mean-squared-error (RMSE) between the learned connectivity matrix $\hat{\boldsymbol{J}}$ and the true matrix $\boldsymbol{J}$. In addition, we compare the efficiency by reporting the running time for each iteration. To be specific, for comparing efficiency, we use the same batch size $100$ for our method and baselines. The report time is the average over $100$ iterations.

$\square$

