# OpenReview forum: "Gradient-Guided Importance Sampling for Learning Binary Energy-Based Models"
_ICLR.cc/2023/Conference — ICLR 2023 poster_

### Official Review · Reviewer_wsgm · 2022-10-17

**Confidence:** 4
**Correctness:** 3
**Technical Novelty And Significance:** 2
**Empirical Novelty And Significance:** 2
**Recommendation:** 5

**Clarity, Quality, Novelty And Reproducibility:**

Overall, the paper is clearly written and well presented. Although ratio matching and scalable sampling of discrete energy based models using gradient information have been explored before, it seems to be a new combination for learning binary energy based models.

**Strength And Weaknesses:**

Strength:

1. The paper is well written and organized.
2. The idea of combining gradient information of the energy function and ratio matching is new.

Weaknesses:

1. As the main idea is a combination of ratio matching and gradient information, there seems to be a lack of ablation study on which one is the main factor for the overall improvement. More specifically, there is no comparison to scalable gradient-based samping method for learning discrete EBMs (e.g., GWG, Grathwohl et al., 2021) in section 5.1, and stochastic ratio matching in section 5.2 and 5.3.

2. The number of samples $s$ seems to be a crucial hyper-parameter and there is no ablation study of it throughout the experiments. It would be better to have it to clarify the choice of $s$ in your experiments. The author reported that the simple random sampling failed when $s=10$ which would be easily saved with more samples.

3. The experiment results are not significant on real data tasks.

**Summary Of The Paper:**

This paper proposed a gradient-guided importance sampling method for learning binary energy based models. The idea is to combine ratio matching with scalable gradient of the energy function for more efficient computation. The paper is clearly written. Numerical experiments show the advantage of the proposed method over alternative baselines.

**Summary Of The Review:**

The paper present a new method for learning binary energy based model that combines ratio matching and scalable gradient-based importance sampling techniques. Overall, the paper is well written and organized. Although ratio matching and gradient-based information for scalable sampling from discrete energy based models have been proposed before, the combination for learning binary energy based model is new. The lack of novelty and significant empricial evidence of the proposed method is the main reason for my recommendation.

---

> ### Author Response · Authors · 2022-11-11
> **Added additional experiments and ablation studies.**
>
> Thank you for the constructive comments.
>
> > As the main idea is a combination of ratio matching and gradient information, there seems to be a lack of ablation study on which one is the main factor for the overall improvement. More specifically, there is no comparison to scalable gradient-based sampling method for learning discrete EBMs (e.g., GWG, Grathwohl et al., 2021) in section 5.1, and stochastic ratio matching in section 5.2 and 5.3.
>
> > The experiment results are not significant on real data tasks.
>
> - Since our main contribution is to improve ratio matching in terms of efficiency and effectiveness, we can directly compare to ratio matching to demonstrate that our proposed gradient-guided importance sampling strategy is more efficient and effective to train binary EBMs. Our comparisons to ratio matching on density modeling with synthetic data (Sec 5.1) and graph generation (Sec 5.2) can demonstrate this. To further support this main claim, in our revision, we additionally added the ratio matching as a baseline in training Ising models (Table 3 in Sec 5.3). The comparison between ratio matching and our RMwGGIS is presented in the following table, further verifying the effectiveness and efficiency of our method.
>
> |Method|`ratio matching`|`RMwGGIS`|
> |----|----|----|
> |log(RMSE)|-3.99|**-4.06**|
> |time/iter|450.7ms|**13.9ms**|
> |Memory|1913MB|**943MB**|
>
> - Note that we currently do not have the implementation of GWG for the experiment in Sec 5.1, and it is non-trivial to implement it in such a short time. Having said this, we have compared with GWG in training Ising models in Sec 5.3 to evaluate the effectiveness and efficiency. Also, we have trained stochastic ratio matching (i.e., RMwRAND in our paper) for graph generation (Sec 5.2) and learning Ising models (Sec 5.3) during the rebuttal period. RMwRAND is not able to converge to a reasonably good result, which aligns with our observations in Sec 5.1. This further verifies that our proposed gradient-guided importance sampling strategy is effective.
> - In terms of real-world tasks, we considered comparing with many recent advanced methods for graph generation (Sec 5.2). In addition, we would like to point out the results on training Ising models (Sec 5.3) is quite convincing. Specifically, the results can directly and fairly compare the performance since the learned energy-based models (specifically, their parameters) are compared to the true connectivity matrix $J$. Thus, a lower RMSE in the results directly means a better learned energy-based model (i.e., a better inferred Ising model). Hence, our added experiment above can provide stronger support for our main claims.
>
> > The number of samples $s$ seems to be a crucial hyper-parameter and there is no ablation study of it throughout the experiments. It would be better to have it to clarify the choice of s in your experiments. The author reported that the simple random sampling failed when $s=10$ which would be easily saved with more samples.
>
> Note that we did not perform extensive tuning for $s$, although tuning it might bring performance improvement. To further show the influence of $s$ and the effectiveness of RMwGGIS over RMwRAND (the ablation model performing random sampling), we added an experiment to train both models on 256-dimensional 2spirals with different $s=5, 10, 50, 100$. The visualization of the learned energy functions is shown in Figure 2 in our revision.
>
> According to the experimental results, our RMwGGIS achieves good density modeling performance that is robust to the selection of $s$. In contrast, RMwRAND fails to capture the data distribution even with $s=100$. This further demonstrates the effectiveness and robustness of our proposed gradient-guided importance sampling strategy.
>
> Thank you for the comments. Please let us know if there are any further questions and discussion.

---

### Official Review · Reviewer_KMq7 · 2022-10-18

**Confidence:** 4
**Correctness:** 4
**Technical Novelty And Significance:** 3
**Empirical Novelty And Significance:** 3
**Recommendation:** 8

**Clarity, Quality, Novelty And Reproducibility:**

This work is overall well-explained and original. I think the method is novel and quite interesting. The experiments are well-done and reproducible.

**Strength And Weaknesses:**

The paper is very well-written and clear. There is a very good description of the background materials and the method itself. There are many experiments to back up their claims.

I feel it could be explained more clearly in the main body that the “advanced version” is the method actually being used. It should be emphasised earlier in the work that the final method does not re-weight the samples according to the importance weights. My understanding is that both “basic” and “advanced” are denoted as “RMwGGIS”, which is confusing.

Are there any experiments showing why the variance of the importance-weighted sampler is better?

Besides on toy-data, can you show that the energy functions learned with your approach are better? e.g. can you show that you can do OOD detection with your models? Is it easier to sample from your learned energy functions compared to those where MCMC is used at training time?

Do you stop_grad through the importance weights? If yes, this should be emphasised.

The top bar notation above equation 9 should be explained in a footnote, and it should be formatted better (e.g. \overline{x}_i, not \overline{x_i})

Can you explain in the main body why you remove the importance sampling? How much bias does this induce in the estimator?

It should be emphasised earlier in the work (e.g. when introducing ratio matching), that this approach doesn’t work well for images.

**Summary Of The Paper:**

This paper proposes a new method to learn discrete EBMs based on ratio-matching. Specifically, they consider the “discrete extension of generalised score matching”. The main disadvantage of this approach is that it is inefficient in time and memory complexity since you need to compute d + 1 energies (d perturbations and x itself), then backprop through it. This paper proposes sampling s << d perturbations instead, and reducing the variance of this estimator via importance sampling. The optimal proposal distribution for importance sampling has a closed form, but is as expensive to compute as the original ratio matching approach. This proposal can be approximated via a first-order Taylor-approximation, following the GWG approach (related work). In practice, the authors dispense with reweighting the ratios by the importance factor (Eq. 11).

**Summary Of The Review:**

The paper is well-written and clear, and the method is novel and interesting. Many of the claims are well-supported by either/both theoretical and empirical justification.

There are two versions of their method being used, and it’s unclear which they refer to in which experiment.

I have some concerns regarding properties of their method. Specifically, experiments that show the bias/variance of the estimator, and experiments on higher-dimensional data showing that the learned energy functions are better (e.g. are the energy functions as easy to sample from at test-time if sampling is not done during training).

---

> ### Author Response · Authors · 2022-11-11
> **Response (Part 2)**
>
> > Can you explain in the main body why you remove the importance sampling? How much bias does this induce in the estimator?
>
> - Thank you for the suggestion. We explained why we remove the importance weights to obtain the advanced version in our revised paper. Basically, during our method development, we first obtained RMwGGIS(basic) with grounded theory, and empirically observed that it is indeed more efficient than the original ratio matching. In addition, as described in Sec 3.3, we experimentally observed that our RMwGGIS(basic) achieves better practical performance than ratio matching. Diving into this, we conjectured that this is because RMwGGIS(basic) focuses on pushing up the energies of neighbors with low energies, thus leading to better optimization in practice as the hard negative mining philosophy (Detailed analysis in Appendix C). Following the hard negative mining perspective, we found that RMwGGIS(basic) gives smaller weights to terms with lower energies in the loss function, which is intuitively less effective from the hard negative mining view (Details in Appendix D). Thus, we further proposed the heuristic RMwGGIS(advanced) by removing the weights in RMwGGIS(basic). Note that, as RMwGGIS(basic), the sampling in RMwGGIS(advanced) (Eq. (11)) still follows the same (approximately) optimal proposal.
> - As in our paper, we have theoretical support for RMwGGIS(basic) and we currently do not have a theoretical result for RMwGGIS(advanced) regarding the bias analysis. Note that both versions are our contributions, and both are verified to be more efficient and more effective than the original ratio matching. RMwGGIS(advanced) is our extended heuristic method and we have emphasized it is heuristics in our revision.
>
> > It should be emphasized earlier in the work (e.g. when introducing ratio matching), that this approach doesn’t work well for images.
>
> As suggested, we have added this clarification to the introduction of ratio matching (Sec 2.2, marked in red). Thank you.
>
> Thank you again for your constructive review. Please let us know if there are any further feedback.
>
>
> [1] Grathwohl, Will, et al. "Oops i took a gradient: Scalable sampling for discrete distributions." International Conference on Machine Learning. PMLR, 2021.

---

> > ### Comment · Reviewer_KMq7 · 2022-11-15
> > **Response**
> >
> > Thanks for your response and adding those changes.
> >
> > I am still concerned about not having an experiment directly computing the variance of your estimator, and comparing it to the variance of the baseline approach. My understanding is that you have experiments which shows your approach does better in the actual modelling task, but not directly computing the variance of your estimator and seeing how it compares to the baseline.
> >
> > Besides this, I am happy to recommend the paper for acceptance and have increased my score.

---

> > > ### Author Response · Authors · 2022-11-16
> > > **Thanks**
> > >
> > > Thank you for raising the score and recommending our paper for acceptance.
> > >
> > > According to Fact 1, if we rigorously follow the optimal proposal to do the importance sampling, the variance would be minimized for sure. In our paper, we further use gradient-based approximation to construct the optimal proposal approximately. We probably cannot directly show how accurate this approximation is since it would be complicated to analyze the error of the gradient-based approximation of the neural network output. To empirically measure the variance, we need to run the estimator for (statistically) significantly many times, which is challenging. Since we connect our method philosophy with hard negative mining, as detailed in Appendix C, we provide an intuitive analysis of the gradient-based approximation in Figure 3, Appendix C. Hope this helps. Thank you.

---

> ### Author Response · Authors · 2022-11-11
> **Response to the questions; revised the paper as suggested; added additional experiments. (Part 1)**
>
> Thank you for your comprehensive review.
>
> > I feel it could be explained more clearly in the main body that the “advanced version” is the method actually being used. It should be emphasized earlier in the work that the final method does not re-weight the samples according to the importance weights. My understanding is that both “basic” and “advanced” are denoted as “RMwGGIS”, which is confusing.
>
> Thanks for this suggestion. As suggested, we have revised our paper to emphasize RMwGGIS(advanced) earlier in the introduction. Also, we would like to point out that both the basic and advance versions are our contributions, and both are verified to be more efficient and more effective than the original ratio matching.
>
> > Are there any experiments showing why the variance of the importance-weighted sampler is better?
> - According to Fact 1, theoretically, the variance associated with the importance-weighted sampler is the minimum. Empirically, as in Sec 5.1 and Figure 4 (Appendix F), we also show that our proposed method significantly outperforms RMwRAND, an ablation model which estimates the original ratio matching objective by randomly sampling. This empirical comparison shows that the variance of the importance-weighted sampler is actually better.
>
> > Besides on toy-data, can you show that the energy functions learned with your approach are better?
>
> Besides the verification of learning better energy function with our approach on synthetic data (Table 1 and 4), we additionally added the ratio matching as a baseline in training Ising models (Table 3 in Sec 5.3), which can directly compare the learned energy-based models since the learned energy-based models (specifically, their parameters) are directly compared to the true connectivity matrix $J$. Thus, a lower RMSE in the results means a better learned energy-based model (i.e., a better inferred Ising model). The comparison between ratio matching and our RMwGGIS is presented in the following table, verifying that our method can lead to a better learned energy-based model as well as improved efficiency in terms of training time and memory usage. We also tried using RMwRAND (the ablation model) on this experiment, it is not able to converge to a reasonably good result.
>
> |Method|`ratio matching`|`RMwGGIS`|
> |----|----|----|
> |log(RMSE)|-3.99|**-4.06**|
> |time/iter|450.7ms|**13.9ms**|
> |Memory|1913MB|**943MB**|
>
>
> > Is it easier to sample from your learned energy functions compared to those where MCMC is used at training time?
>
> Our approach is focusing on training/learning energy-based model more effectively and efficiently. For sampling, as other MCMC based training method, we still use MCMC sampling methods to sample from the learned energy function. We can also use the recently proposed gradient-based sampling method [1] to sample from the learned energy function more efficiently. We did not include such sampling technique since we mainly focusing on learning EBM instead of sampling from learned EBM.
>
> > Do you stop_grad through the importance weights? If yes, this should be emphasized.
>
> Yes. We have emphasized this in our revision. Thanks for the suggestion.
>
> > The top bar notation above equation 9 should be explained in a footnote, and it should be formatted better (e.g. \overline{x}_i, not \overline{x_i})
>
> We have explained it at the beginning of Sec 2.2. As suggested, we use the better format in our revision. Thank you!
>
> -----> the remaining response is in the following part 2 ----->

---

### Official Review · Reviewer_D4Cv · 2022-10-24

**Confidence:** 3
**Correctness:** 3
**Technical Novelty And Significance:** 3
**Empirical Novelty And Significance:** 3
**Recommendation:** 6

**Clarity, Quality, Novelty And Reproducibility:**

In terms of clarity, I think the paper is clearly written and easy to follow, so I do not have much complaints about the presentation, apart from the reason why the advanced version performs better empirically. In terms of novelty, although the idea is simple, it seems to be a novel contribution to the best of my knowledge. I have briefly checked the math, which seems to be good. For reproducibility, the author provides the settings for the experiments in the appendix. Since the author does not provide the code, I cannot say 100% the experiments are reproducible.

**Strength And Weaknesses:**

## Strength

The paper is clearly written and easy to follow. The author also explains the intuition behind the proposed importance sampling approach in section 3.3, which help the understanding of the paper. Empirically, the author considers three tasks and includes many baselines in graph generation task. I briefly checked the math, which seems to be correct.

## Weakness
Theoretically, I am curious about why eq.11 has a better performance than eq.6? Eq.11 is not a proper objective anymore, which means it does not correspond to a valid discrepancy. How does this ensure that the model can converge to the correct distribution? In addition, from theorem 1, the importance proposal should be the optimal one (at least approximately). Why the advanced version is better?

Empirically, I think other than the graph generation task, more baselines should be considered. At the moment, only ratio matching is used for synthetic data and GWG and Gibbs are used for the Ising model. More baselines can further support the claims made by the paper.

**Summary Of The Paper:**

This paper proposed an importance sampling approach to alleviate the computation and memory complexity of ratio matching for training binary EBM. Specifically, the author first rewrites the ratio-matching objective as an expectation of the energy difference and then showed the form of optimal importance proposal to reduce the variance of the objective. In the end, the author showed that the optimal importance proposal can be approximated by using the Taylor expansion following the ideas from Gibbs-with-gradient (GWG) sampler.

Empirically, the author evaluates the proposed approach using synthetic data, graph generation, and ising model. In general, it achieves better results in the graph generation task. Although it performs worse than GWG, it is much faster in terms of training.

**Summary Of The Review:**

The author proposed an improved version of ratio matching for training binary EBM by using importance sampling. In general, the paper is written clearly and does a good job of explaining the intuition of the proposed method. I am also curious why there is a discrepancy between the theory and empirical evaluations such that the advanced version performs better than the "correct" version.

---

> ### Author Response · Authors · 2022-11-11
> **Both versions are our contributions; included additional empirical comparison.**
>
> Thank you for your constructive comments.
>
> > I am curious about why Eq. (11) [RMwGGIS(advanced)] has a better performance than Eq. (6) [RMwGGIS(basic)]?
> - As in our paper, we have theoretical support for RMwGGIS(basic) and we currently do not have a theoretical analysis for RMwGGIS(advanced). Note that both versions are our contributions, and both are verified to be more efficient and more effective than the original ratio matching. RMwGGIS(advanced) is our extended heuristic method, which is obtained as follows. We have emphasized it is heuristics in our revision.
> - During our method development, we first obtained RMwGGIS(basic) with grounded theory, and empirically observed that it is indeed more efficient than the original ratio matching. In addition, as described in Sec 3.3, we experimentally observed that our RMwGGIS(basic) achieves better practical performance than ratio matching. Diving into this, we conjectured that this is because RMwGGIS(basic) focuses on pushing up the energies of neighbors with low energies, thus leading to better optimization in practice as the hard negative mining philosophy (Detailed analysis in Appendix C). Following the hard negative mining perspective, we found that RMwGGIS(basic) gives smaller weights to terms with lower energies in the loss function, which is intuitively less effective from the hard negative mining view (Details in Appendix D). Thus, we further proposed the heuristic RMwGGIS(advanced) by removing the weights in RMwGGIS(basic). Note that, as RMwGGIS(basic), the sampling in RMwGGIS(advanced) (Eq. (11)) still follows the same (approximately) optimal proposal.
>
> > More baselines can further support the claims made by the paper.
>
> - We agree that it would make our submission stronger to include more baselines. But we would like to point out that it needs a lot of efforts to reimplement/reproduce more baselines in our experiments, since many baselines are largely different from our method implementation and do not have public code.
>
> - On the other hand, we have made efforts to include the most necessary and required baselines, thus supporting our main claims well. Particularly, we compared with ratio matching in both Sec 5.1 and 5.2, since our main contribution is to improve ratio matching in terms of efficiency and effectiveness. We also compared with the classic method that trains discrete EBM with persistent contrastive divergence (i.e., EBM(Gibbs) in Sec 5.3), and the most recent EBM(GWG) method, for more thorough empirical comparison.
> - In our revision, we additionally added the ratio matching as a baseline in training Ising models (Table 3 in Sec 5.3). Besides that, we added an experiment to train both RMwGGIS and RMwRAND on 256-dimensional 2spirals with different $s=5, 10, 50, 100$. The visualization of the learned energy functions is shown in Figure 2 in our revision. These experimental results further demonstrates the effectiveness and robustness of our proposed gradient-guided importance sampling strategy.
>
>
> > Reproducibility.
>
> We will publish our code with our paper for reproducibility.
>
> Thank you again for your comments and suggestions. Please let us know if there are any further feedback. Thank you!

---

### Official Review · Reviewer_pBMS · 2022-10-25

**Confidence:** 4
**Correctness:** 4
**Technical Novelty And Significance:** 3
**Empirical Novelty And Significance:** 3
**Recommendation:** 8

**Clarity, Quality, Novelty And Reproducibility:**

The paper is very clearly written and it is a pleasure to read.

The proposed method is very novel.

The work is of a high quality.

Can you generalize your method by flipping multiple sites?

**Strength And Weaknesses:**

Strengths:

(1) The proposed method is novel, simple and useful.

(2) The experiments illustrate the effectiveness of the proposed method.

(3) The paper clearly discusses the limitation with respect to binary images.

Weaknesses:

Methods based on importance sampling may not work well for high dimensional problem.

**Summary Of The Paper:**

This paper studies the problem of learning binary energy-based model using ratio matching method. The paper proposes to use importance sampling to improve the time and memory efficiency of the ratio matching method, where the proposal distribution is obtained based on the gradient of the energy function with respect to the binary variables. The effectiveness of the proposed method is demonstrated on synthetic and real datasets.

**Summary Of The Review:**

The paper proposes a simple and effective method for learning binary energy-based model.

---

> ### Author Response · Authors · 2022-11-11
> **Discussion to the open questions; added additional experiments.**
>
> Thank you for recognizing our work to be novel, simple, and effective.
>
> > Methods based on importance sampling may not work well for high dimensional problem.
>
> - Thanks for raising this interesting consideration. Yes, it is known that the variance of the importance sampling grows as the data dimension increases. We might need to use specifically designed importance sampling strategies for high-dimensional data, where the normal importance sampling does not work. There are several existing works studying the importance sampling methods in high dimensions [1,2], which could be helpful for this case.
>
> - On the other hand, for our proposed approach, we have verified the ability of scaling to high-dimensional problems in Sec 5.3, where the Ising model is performed on 625-dimensional data. We also additionally include ratio matching as a baseline in this experiment in our revision. This demonstrates that our method can scale to high-dimensional problem in practice. We leave the consideration for extremely high dimensional problem, where the normal importance sampling does not work, as future research.
>
> > Can you generalize your method by flipping multiple sites?
>
> - Great question! We describe our preliminary thoughts on this open question here. To make it simple, let us assume that we want to flip 2 dimensions. In this case, we can still use Taylor series like Eq. (7) to approximate the energy associated with the flipped data. In addition, a similar equation as Eq. (8) still exists for estimating the energies of all possible flipped data ($C(n,2)=O(n^2)$). Specifically, the left multiplier in Eq. (8) would become a matrix instead of a vector. Flipping more dimensions can also be similarly approximated. Therefore, the gradient-guided approximation still holds for the case of flipping multiple dimensions.
>
> - The bottleneck is that only minimizing the ratio matching objective, where one-dimensional flipping is applied as shown in Eq. (3), have been proved to be a consistent estimator [3,4]. Thus, we currently cannot easily extend the gradient-guided importance sampling idea to ratio matching by flipping multiple sites. But the above analysis for flipping multiple sites might be applicable for other problems.
>
> Thank you again for the positive comments and raising this interesting discussion. Looking forward to any further discussion.
>
>
> [1] Scharth, Marcel, and Robert Kohn. "Particle efficient importance sampling." Journal of Econometrics 190.1 (2016): 133-147.
>
> [2] Richard, Jean-Francois, and Wei Zhang. "Efficient high-dimensional importance sampling." Journal of Econometrics141.2 (2007): 1385-1411.
>
> [3] Hyvärinen, Aapo. "Some extensions of score matching." Computational statistics & data analysis 51.5 (2007): 2499-2512.
>
> [4] Lyu, Siwei. "Interpretation and generalization of score matching." Proceedings of the Twenty-Fifth Conference on Uncertainty in Artificial Intelligence. 2009.

---

### Decision · Program_Chairs · 2023-01-20

**Decision:**

Accept: poster

**Justification For Why Not Higher Score:**


The method is a combination of existing techniques: Gibbs-with-gradient and ratio matching.

The empirical comparison part is weak:

- The method is limited that it cannot be applied for binary image data, which is the major tasks in literature.
- The competitors are not comprehensive in synthetic data and Ising model

**Justification For Why Not Lower Score:**


I personally would like to reject this paper. However, most of the reviewers think the paper is valuable to be published.

Only one reviewer (Reviewer wsgm) pointed out the novelty and significance issue.



**Metareview: Summary, Strengths And Weaknesses:**


In this paper, the authors combined the Gibbs-with-gradient sampler and ratio matching for binary energy-based model training. The authors showed the benefits of the proposed method on several datasets.

Although the paper is well-written and easy-to-follow, and most of the reviewers recognize the corresponding signficance. I have mixed feeling of this paper.

As Reviewewr wsgm pointed out, the novelty and signficance of the proposed method is not enough. The success of the proposed method is also expactable, as the GWG and ratio matching are both working in practice.

Meanwhile, the method is limited that cannot be applied to binary image modeling task, which is the major task in the other alternatives in the literature, e.g., GWG. This limitation largely restricted the application of the method.

The empirical part is also weak, in the sense of lacking of strong baselines, especially in synthetic data and Ising model.

**Note From Pc:**

if the above contains the word "oral" or "spotlight" please see: "oral" presentation means -> notable-top-5% and "spotlight" means -> notable-top-25%. As stated in our emails, we are disassociating presentation type from AC recommendations